

# Comparison of box counting and correlation dimension methods in well logging data analysis associate with the texture of volcanic rocks

**D. Mou**[1] **and Z. W. Wang**[2]

[1]School of Mathematics and Statistics, Beihua University, Jilin, China
[2]College of Geo-exploration Science and Technology, Jilin University, Changchun, China

Received: 15 October 2014 – Accepted: 25 May 2015 – Published: 12 July 2016

Correspondence to: D. Mou (mudan-main@163.com)

Published by Copernicus Publications on behalf of the European Geosciences Union & the American Geophysical Union.



**NPGD**

doi:10.5194/npg-2014-85

**Comparison of box counting and correlation dimension methods**

D. Mou and Z. W. Wang

## Abstract

We have developed a fractal analysis method to estimate the dimension of well logging curves in Liaohe oil field, China. The box counting and correlation dimension are methods that can be applied to predict the texture of volcanic rocks with calculation the fractal dimension of well logging curves. The well logging curves are composed of gamma ray (GR), compensated neutron logs (CNL), acoustic (AC), density (DEN), Resistivity lateral log deep ($R_{LLD}$), every curve contains a total of 6000 logging data. The dimension of well logging curves are calculated using box counting and correlation algorithms respectively. It is shown that two types of dimension of CNL, DEN and AC have the same average value. The box counting dimension of volcanic lava is lower than the pyroclastic rock obviously. The majority of correlation dimension of volcanic lava is lower than the pyroclastic rock, but a small amount of correlation dimension of volcanic lava is equal to the pyroclastic rock. It is demonstrated that the box counting dimension is more suitable for predicting the texture of volcanic rocks. Applications to logging data, A well show the relationship between the fractal dimension and the texture of volcanic rock in certain depth.

## 1 Introduction

The geological processes that the earth experienced is considered as a non-linear process, which led to strong heterogeneity in rock lithology, permeability and porosity distribution (Vadapalli et al., 2014). The heterogeneity provides premise and feasibility that the study of fractal theory is applied to geophysics (Flores-Marquez et al., 2012). To solve the problems in geology, many scholars investigate the fractal characteristics of well logging data, and make a great progress (Li, 2003). A lot of theoretical research about the fractal characteristic of well logging curves has been done, which mainly focus on self-similarity, fractal interpolation, fractal correction multi-fractal analysis and calculating method of fractal dimension (Dimri, 1998; Tarquis et al.,

Discussion Paper | Discussion Paper | Discussion Paper | Discussion Paper

2014; Wang et al., 2014). Besides, the geological information in well logging data was quantitative interpreted using fractal dimension of well logging curves. Furthermore, fractal dimension has a wide application in the prediction of fracture, reservoir heterogeneity, sedimentary facies analysis and rock pore structure description.

Fractal characteristics of well logging curves reflect the volcanic rock lithology, physical property, cracks and fluid nature comprehensively. The value of well logging parameters often reflect the mineral composition of rocks, but not reflect the texture of volcanic rocks (Li, 2005). However, Fractal dimension is aimed to describe the complexity of logging curves, which is independent of well logging parameters. The texture of volcanic rocks refer the crystallization, particle size and morphology as well as the relationship between these substances, which will change the complexity of logging curves (Zhou et al., 2014). Therefore, the fractal dimension can quantitative described the complexity of logging curves in shape and amplitude.

In this paper, we first apply the box counting and correlation dimension methods to calculation the fractal dimension of well logging curves (Liebovitch et al., 1989). Then we compare the results of the two types methods. Lastly, we discuss the relationship between fractal dimension of well logging curves and the texture of volcanic rocks.

## 2  Box counting dimension

Fractal dimension is a useful concept describing the complexity of natural objects. There are many algorithms to calculate the fractal dimension. The box counting dimension is widely used in practice, and is easy to implement (Saa et al., 2007; Li et al., 2009; Borodich et al., 2010).

In box counting dimension method, a planar set $S$ are divided into square boxes, which are a square $\delta$ on a side and $S$ is covered with square boxes, and the number of square boxes of a certain size is counted to see how many of them are necessary to cover $S$ completely. When $\delta \rightarrow 0$, the total area covered with square boxes will

**NPGD**

doi:10.5194/npg-2014-85

**Comparison of box counting and correlation dimension methods**

D. Mou and Z. W. Wang

converge to the measure of $S$. This can be expressed as

$$D_B = \lim_{\delta \to 0} \left( \log N_\delta(S) / \log(1/\delta) \right),$$

where $N_\delta(S)$ is the total number of boxes of size $\delta$, that required to cover the $S$ entirely. In practice, the box-counting algorithm estimates fractal dimension by counting the number of boxes required to cover $S$ for several box sizes, and fitting a straight line to the log-log plot of $N_\delta(S)$ vs. $\delta$. The slope of the least square best fit straight line is taken as the box-counting dimension $D_B$.

Well logging curves are non-regular plane curves, which was connected with measured value for each sample points. To satisfy point-by-point calculation, we adopt the window technique, which actually was analyzed for well logging curves from one window length.

We take the Y well, 1953–1954 m sections with a total of 9 sample points for example (Fig. 1), and research the calculation procedure for box-counting dimension of well logging curves.

1. Where the window is 1 m in length, the well logging curves in one window length were mapped into a square $L = 1$ on a side. Where the starting depth of well logging curves is $H_{start}$, the ending depth is $H_{end}$, and the depth of sample point is $H_{log}$. The maximum value of sample point is $V_{max}$, the minimum value is $V_{min}$, and the log value of each sample point is $V_{log}$, so the coordinates of each sample point in the plane:

$$V_x = L(V_{log} - V_{min})/V_{max} - V_{min},$$
$$V_y = L(H_{log} - H_{start})/V_{end} - V_{start}. \tag{1}$$

2. We calculate the equations of sampling points of two adjacent lines. The line equation as follows can be derived from the two-point form of straight line equation:

$$y = V_{y(i)} + (V_{y(i+1)} - V_{y(i)})(x - V_{x(i)})/V_{x(i+1)} - V_{x(i)}(V_{x(i)} \le x \le V_{x(i+1)}), \tag{2}$$

Discussion Paper | Discussion Paper | Discussion Paper | Discussion Paper | Discussion Paper |

**NPGD**

doi:10.5194/npg-2014-85

**Comparison of box counting and correlation dimension methods**

D. Mou and Z. W. Wang

where the number of sampling points in one window is 9, so $i = 1, 2, \ldots, 8$.

3. We adopt grid processing for planar graph. Suppose to the grid points in horizontally or vertically are equal to $M$. The coordinates for each split point are as follows:

$$X_{\text{grid}(i)} = iL/M,$$
$$Y_{\text{grid}(i)} = iL/M \quad (i = 0, 1, \ldots, M). \tag{3}$$

So the planar graph is decomposed into $M \times M$ small squares that are the boxes of $\delta = L/M$ side length. The following equations are used to describe boxes:

$$x_i = X_{\text{grid}(i)},$$
$$y_i = Y_{\text{grid}(i)} \quad (i = 0, 1, \cdots, M). \tag{4}$$

4. We solve the Eqs. (2) and (4), the solution of equations are the points at which well logging curves intersects boxes, and $W$ is called the number of intersection points. There are two kinds of situation that calculate the number of boxes: in case the intersection points do not include the boundary points, then $N_\delta = W - 1$ is called the number of boxes. In another case, the intersection points include the boundary points, then $N_\delta = W - 2$ is called the number of boxes.

5. We conduct $K$ times grid processing for planar graph ($K = 100$), and obtain a total of 100 groups data of $(\delta_{(i)}, N_{\delta(i)})$ ($i = 1, 2, \ldots, K$). Then we obtain $(\log \delta_{(i)}, \log N_{\delta(i)})$ by logarithmic transformation of $(\delta_{(i)}, N_{\delta(i)})$. Afterwards, we do the least squares line fitting for $(\log \delta_{(i)}, \log N_{\delta(i)})$, the straight line equation $y = kx + b$ is derived. The result is shown in Fig. 2.

## 3  Correlation dimension

Correlation dimension represent the association degree of data series. On the basis of the embedding theory and the idea of phase space reconstruction, Grassberger

**NPGD**

doi:10.5194/npg-2014-85

**Comparison of box counting and correlation dimension methods**

D. Mou and Z. W. Wang

and Procaccia put forward the algorithm for calculating correlation dimension directly from the time series, which is known as G-P algorithm. The method of phase space reconstruction expand the data series that observed at different times within a certain time to a vector space, that is, phase space. Well logging data is associated with the formation of physical properties, which is measured with the same depth sampling interval. So we first expand the well logging data to phase space, then calculate its correlation dimension.

We take the Z well, 2162–2168 m sections with a total of 49 sample points for example, and describe the calculation procedure for correlation dimension of well logging curves.

1. The $n$ dimension phase space is established by well logging data. Where $M$ is the number of logging data, when constructing the first vector, take $N$ data starting at the first logging data, so let every logging data be the value of this vector and the rest may be deduced by analogy, we expand the $M$ logging data into $K$ vectors, and the phase space is formed. The every vector called $\overline{y}_i$ ($i = 1, 2, \ldots, K$) can be obtained as:

$$\begin{cases} \overline{y_1} = (x_1, x_2, \cdots x_N) \\ \overline{y_2} = (x_2, x_3, \cdots x_{N+1}) \\ \vdots \\ \overline{y_i} = \left(x_i, x_{i+1}, \cdots x_{N+i-1}\right) \\ \vdots \\ \overline{y_k} = \left(x_K, x_{K+1}, \cdots x_{N+K-1}\right) \end{cases}. \tag{5}$$

Discussion Paper | Discussion Paper | Discussion Paper | Discussion Paper | Discussion Paper |

**NPGD**

doi:10.5194/npg-2014-85

**Comparison of box counting and correlation dimension methods**

D. Mou and Z. W. Wang

2. The Euclidean distance between any two vectors in phase space is calculated, which is given as:

$$R_{i,j} = \left| \overline{y_i} - \overline{y_j} \right| = \sqrt{\sum_{k=1}^{N} (x_{k+i-1} - x_{k+j-1})^2},$$
(6)

where $\overline{y_i}$ and $\overline{y_j}$ are the two vectors.

3. In practice, for any given scale $\varepsilon$, the distance number is $N(\varepsilon)$, which satisfy with the condition $R_{i,j} < \varepsilon$. Whereas the total distance number of any two vectors is $K^2$, the proportion of the number of qualified distance in total distance at different scale is calculated, also called the $C(\varepsilon)$:

$$C(\varepsilon) = \frac{N(\varepsilon)}{K^2}.$$
(7)

4. We take the logarithm of $\varepsilon$ and $C(\varepsilon)$, as a result of fitting the log-log straight, the slope of which is called correlation dimension. The figure is showed in Fig. 3. The correlation dimension can be obtained as:

$$D = \lim_{\varepsilon \to 0} \frac{\log C(\varepsilon)}{\log \varepsilon}.$$
(8)

5. Non-scale interval refers to a scope with in-variance of scales, subjects within that range has self-similar properties, that is fractal features. When we calculate the correlation dimension, scaleless interval can be defined as the straight line of log-log curve. Where the dimension of phase space is $N = 3, 5, \ldots, 25$, we increase the dimension of the phase space until the correlation dimension is no longer increases with $N$. Then the least square linear fit is conducted for the straight part of the curve. The slope value is correlation dimension of logging curves. As shown

**NPGD**

doi:10.5194/npg-2014-85

**Comparison of box counting and correlation dimension methods**

D. Mou and Z. W. Wang

Discussion Paper | Discussion Paper | Discussion Paper | Discussion Paper |

in Fig. 3a, the larger $N$, the distance of adjacent $\log C(\varepsilon) - \log \varepsilon$ is smaller, which are parallel in the scaleless areas indicating correlation dimension is gradually converging.

## 4   Results and discussion

The study area is located in the eastern depression in Liaohe Basin, China. The lithology in this area is mainly the intermediate and basic volcanic rock. Classification standard was formulated based on the difference of the log response of volcanic rock, which consisted of basalt, non-compacted basalt, trachyte, non-compacted trachyte, gabbro and diabase (Fig. 4). The logging response of volcanic rock is shown in Table 1.

The program for calculating fractal dimension of well logging curves was implemented in Matlab2012a by using box-counting and correlation dimension methods. We sort out five logging parameters including GR, $R_{\text{LLD}}$, AC, DEN and CNL, which are influenced by the texture of volcanic rocks. Then we calculate the box-counting and correlation dimension of well logging curves with a total of 6000 logging data. In this paper, the window is 1 m in length, the sampling rate in one window is 9 and the times of grid processing is 100. The box-counting dimension of logging curves of volcanic rocks is shown in Table 2, and correlation dimension is shown in Table 3.

For the purpose of investigate the relationship between fractal dimension and texture of volcanic rocks, the texture characteristic of volcanic rocks was analyzed based on core data of geology. Seen from Tables 1 and 2, the results are as follows:

1. $D_{\text{DEN}}$, $D_{\text{AC}}$ and $D_{\text{CNL}}$ are called the fractal dimension of DEN, AC and CNL, they have same value for one type of volcanic rock. In addition, $D_{\text{RLLD}}$ is called the fractal dimension of RLLD, which is higher than $D_{\text{AC}}$, $D_{\text{DEN}}$ and $D_{\text{CNL}}$, and $D_{\text{GR}}$ is called the fractal dimension of GR, which is lower than $D_{\text{AC}}$, $D_{\text{DEN}}$ and $D_{\text{CNL}}$.

2. We research the distribution regularity of fractal dimension of volcanic rocks. The more complicate of the texture of volcanic rock, the higher is the fractal dimension.

Discussion Paper | Discussion Paper | Discussion Paper | Discussion Paper | Discussion Paper |

**NPGD**

doi:10.5194/npg-2014-85

**Comparison of box counting and correlation dimension methods**

D. Mou and Z. W. Wang

The fractal dimension of volcanic lava is lower than the fractal dimension of pyoclastic rock. Furthermore, this paper describes the application of fractal dimension to distinguish the super-gene rock from sub-volcanic rock, or separate plutonic rock from volcanic lava, but the effect is not obvious.

To validate the accuracy of the box counting and correlation methods, we calculate the box-counting and correlation dimension for A well, 3250–3570 m sections, and predict the texture of volcanic rock in corresponding formation depth. Then we contrast the predicting result with the lithology of core data, the corresponding rate of box-counting method reaches 90 %, whereas it is lower for the correlation dimension, which corresponding rate is 86.5 %. As shown in Fig. 5, the well logging curves is a fractal system with self-similar structure, which degree of complexity can be represented quantitatively by using fractal dimension. Lastly, the result indicates box counting dimension of well logging curves is suited to predict the texture of volcanic rocks.

## 5   Conclusions

In this paper, our study has illustrated well logging curves a fractal system with self-similar structure, which degree of complexity can be represented quantitatively by using fractal dimension. Moreover, the $D_B$ values of well logging curves have been calculated by box counting method, and the $D_C$ values have been calculated by applying the correlation method. The range of box sizes chosen and the times of grid processing are crucial in determining $D_B$ values. In addition, we have chosen logging parameters of GR, $R_{LLD}$, AC, DEN and CNL with a total of 6000 logging data to calculate their fractal dimension. On the basis of calculating their fractal dimension, it has been concluded that box counting dimension is suited for predicting the texture of volcanic rocks. However, we found that the effect of application of fractal dimension to distinguish the super-gene rock from sub-volcano rock, or separate plutonic rock from volcanic lava is not obvious.

# NPGD

doi:10.5194/npg-2014-85

**Comparison of box counting and correlation dimension methods**

D. Mou and Z. W. Wang

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

Comparison of box counting and correlation dimension methods

D. Mou and Z. W. Wang

**Table 1.** Log response of volcanic rock in eastern depression, Liaohe Basin.

| Lithology | $R_{LLD}(\Omega\,m)$ | AC($\mu$s ft$^{-1}$) | CNL(%) | DEN(g cm$^{-3}$) | GR(API) |
|---|---|---|---|---|---|
| Non-compacted basalt | 10–20 | 60–80 | 24–36 | 2.3–2.6 | 40–56 |
| basalt | 8–12 | 56–72 | 24–29 | 2.6–2.8 | 28–40 |
| Non-compacted trachyte | 42–80 | 68–80 | 16–24 | 2.1–2.3 | 152–160 |
| Trachyte | 44–2000 | 52–60 | 4–8 | 2.3–2.6 | 140–155 |
| Gabbro | 698–8000 | 48–52 | 13–16 | 2.6–2.7 | 60–80 |
| Diabase | 128–800 | 52–64 | 12–18 | 2.6–2.7 | 30–45 |

**Table 2.** The box-counting dimension of logging curves of volcanic rocks.

| Texture of volcanic rock | Box counting dimension of logging curves | | | | | Average dimension |
|---|---|---|---|---|---|---|
| | $D_{RLLD}$ | $D_{AC}$ | $D_{CNL}$ | $D_{DEN}$ | $D_{GR}$ | $D_B$ |
| Lava | 1.224 | 1.123 | 1.113 | 1.147 | 1.092 | 1.140 |
| Pyroclastic lava | 1.307 | 1.265 | 1.226 | 1.292 | 1.118 | 1.253 |
| Pyroclastic rock | 1.356 | 1.348 | 1.353 | 1.230 | 1.248 | 1.321 |
| Sink-pyroclastic rock | 1.132 | 1.103 | 1.092 | 1.085 | 1.050 | 1.094 |
| Super-gene rock | 1.208 | 1.129 | 1.143 | 1.171 | 1.133 | 1.157 |
| Plutonic rock | 1.211 | 1.133 | 1.161 | 1.168 | 1.123 | 1.159 |

NPGD

doi:10.5194/npg-2014-85

Comparison of box counting and correlation dimension methods

D. Mou and Z. W. Wang

Discussion Paper | Discussion Paper | Discussion Paper | Discussion Paper

NPGD

doi:10.5194/npg-2014-85

Comparison of box counting and correlation dimension methods

D. Mou and Z. W. Wang

**Table 3.** The correlation dimension of logging curves of volcanic rocks.

| Texture of volcanic rock | Correlation dimension of logging curves | | | | | Average dimension |
|---|---|---|---|---|---|---|
| | $D_{RLLD}$ | $D_{AC}$ | $D_{CNL}$ | $D_{DEN}$ | $D_{GR}$ | $D_C$ |
| Lava | 1.214 | 1.115 | 1.123 | 1.136 | 1.095 | 1.137 |
| Pyroclastic lava | 1.312 | 1.271 | 1.232 | 1.289 | 1.117 | 1.244 |
| Pyroclastic rock | 1.345 | 1.322 | 1.351 | 1.297 | 1.248 | 1.313 |
| Sink-pyroclastic rock | 1.128 | 1.101 | 1.089 | 1.083 | 1.067 | 1.094 |
| Super-gene rock | 1.254 | 1.123 | 1.157 | 1.182 | 1.133 | 1.170 |
| plutonic rock | 1.246 | 1.132 | 1.153 | 1.178 | 1.139 | 1.170 |

# NPGD

doi:10.5194/npg-2014-85

**Comparison of box counting and correlation dimension methods**

D. Mou and Z. W. Wang

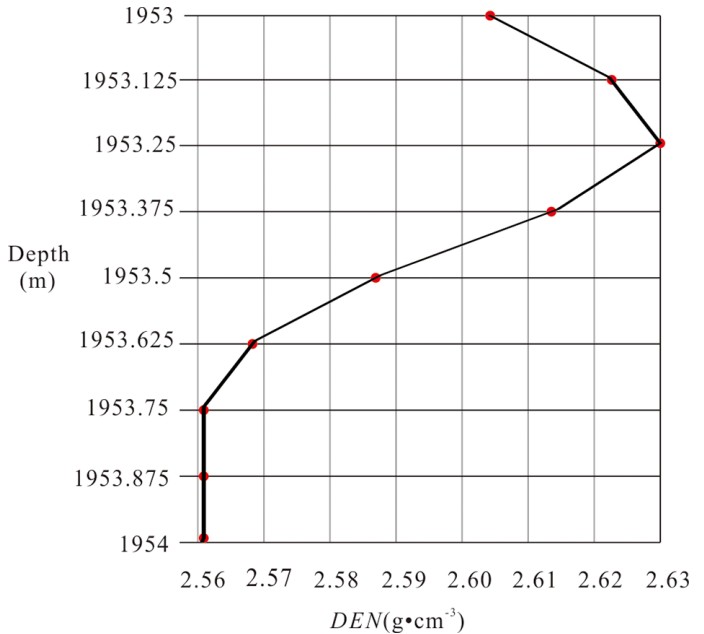

**Figure 1.** The calculation of box-counting dimension of DEN curve.

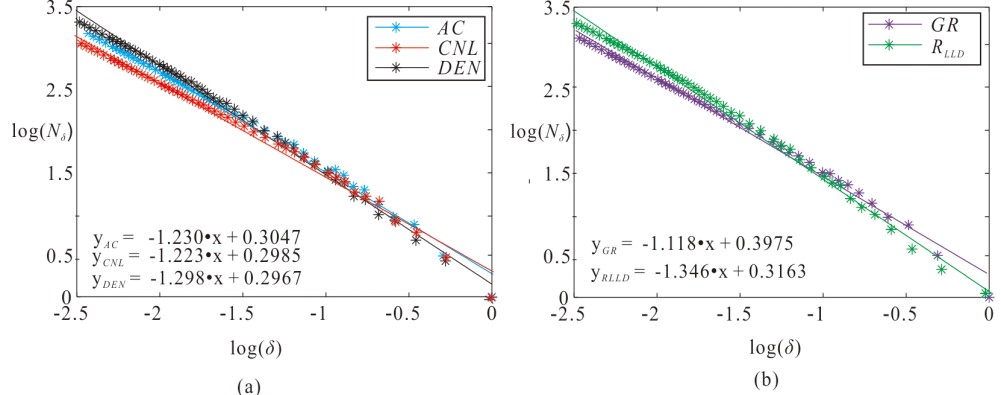

Figure 2. The diagram of calculating box counting dimension.

# NPGD

doi:10.5194/npg-2014-85

**Comparison of box counting and correlation dimension methods**

D. Mou and Z. W. Wang



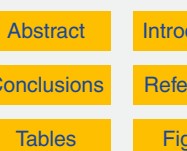
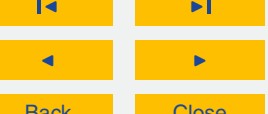
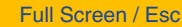

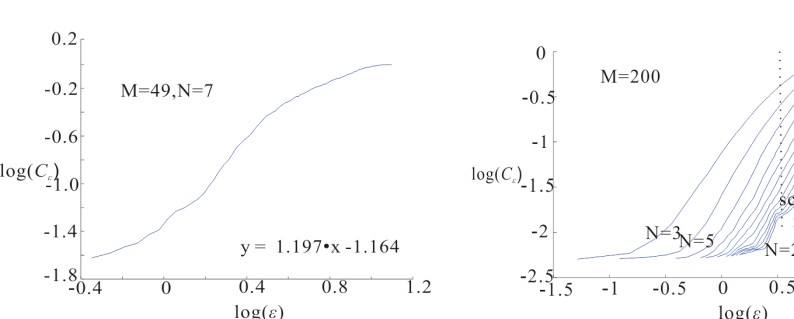

**Figure 3.** Determination of fractal scaleless range.

**NPGD**

doi:10.5194/npg-2014-85

**Comparison of box counting and correlation dimension methods**

D. Mou and Z. W. Wang

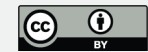

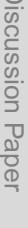

**NPGD**

doi:10.5194/npg-2014-85

**Comparison of box counting and correlation dimension methods**

D. Mou and Z. W. Wang

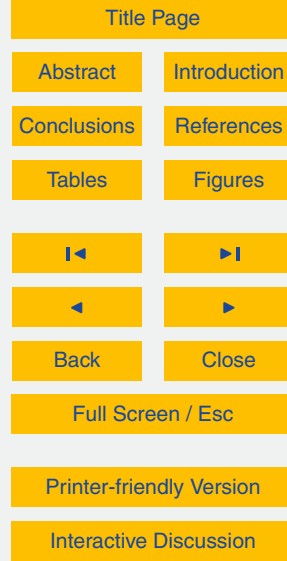

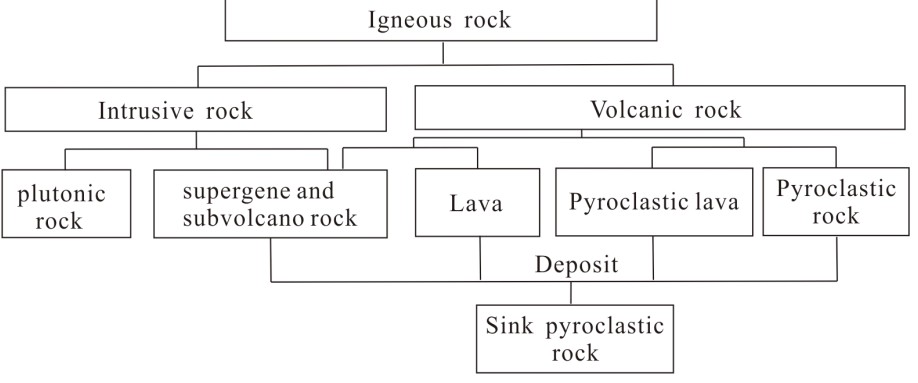

**Figure 4.** The diagram of the texture of volcanic rocks.

# NPGD

doi:10.5194/npg-2014-85

**Comparison of box counting and correlation dimension methods**

D. Mou and Z. W. Wang

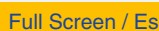

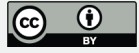

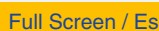

**Figure 5.** Well logging interpretation result from A Well, 2300–2580 m section.