# Peer review of "Comparison of box counting and correlation dimension methods in well logging data analysis associate with the texture of volcanic rocks"

_Nonlinear Processes in Geophysics, 2014_

## Referee Comment (RC1) · Anonymous Referee #1 · 23 Aug 2016

The manuscript entitled "Comparison of box counting and correlation dimension methods in well logging data analysis associate with the texture of volcanic rocks" by Mou and Wang proposes to compare various methods to estimate the fractal dimension of borehole logs in order to estimate the texture of volcanic rocks.

The article cannot be published before major revision. It is very hard to follow the work proposed here. Nevertheless, from my understanding results are significantly new to be considered for publication in Nonlinear Processes in Geophysics journal.

General comments:

The paper needs a section to introduce the dataset: well logging data. Where does it come from? A map of the borehole would be also very interesting. If it has been already published elsewhere, a short description is enough but a citation is required. Page 8 lines 5-9 should be enhanced and added in a new section. What is A well? Z well?

Example of figure 1 is not really useful. Please add some "box" in the graph to help the reader or remove it.

Another figure explaining the correlation dimension computation would also be helpful.

Page 8 line 10-17: I do not understand the data manipulation. How many borehole do you have? What is the total thickness? The spatial sampling? Why using a window? How many records do you have in a window? Do you compute a fractal dimension per borehole? per geological unit? Per texture? What is the "time of grid processing"? How many points do you use to estimate a single fractal dimension?

The section "results and discussion" is very confusing. Please separate the data analysis (and data manipulation), the notation and put it in a "method" section. The results and discussion can stay in this last section.

Page 8 line 5-14: it is a very good idea to predict the texture from the fractal dimension. Please explain it into detail because the reader cannot understand it.

Since the difference between fractal dimension of lava and pyroclastic rock is very small, it is important to estimate the uncertainties to discuss the relevance of this difference.

Minor comments:

In my opinion, the use of such acronyms should be better avoid because it make the paper difficult to follow: gamma ray (GR), compensated neutron logs (CNL), acoustic (AC), density (DEN), Resistivity lateral log deep (RLLD).

Page 1 line 15: Last sentence of the introduction has to be rephrased "Applications to logging data, A well show the relationship between the fractal dimension and the texture of volcanic rock in certain depth."

Page 3 line 15: incorrect citation to (Liebovitch et al., 1989). Please cite it in line 21.

Page 5 line 24: "Grassberger and Procaccia" are not in the references

---

## Referee Comment (RC2) · Anonymous Referee #2 · 13 Sep 2016

Comments on "Comparison of box-counting and correlation dimension methods in well logging data analysis associate with the texture of volcanic rocks" by D. Mou and Z. W. Wang

The authors have come a long way in making their paper easier to comprehend. The original submission did not make much sense but now the reader can follow largely what the authors have done. Nevertheless, many little problems continue to make the reading difficult. Could an English-speaking guide still provide further help? For example, further corrections or clarifications could be requested for where I put question

marks in the following paragraph: "Well logging curves are non-regular plane curves (?), which was (?) connected with measured value (?) for each sampling points (?). To satisfy point-by-point calculation we adopt the window technique which actually was analysed for well logging curves from one window length (?).

In Fig. 3a, a correlation dimension is being estimated for M = 49 and N = 7 where M is number of logging data (?) and N is number of data taken at the first logging data (?). The solution is written as y = 1.197*x -1.164. This best-fitting straight line provides a poor fit to the data shown on the diagram.

In Table 2: "Texture" probably means "Type". Why is pyroclastic rock different from pyroclastic lava? What is "sink-pyroclastic" rock?